# π-phase modulated monolayer supercritical lens

Fei Qin[1,7], Boqing Liu[2,7], Linwei Zhu[3], Jian Lei[1], Wei Fang[1], Dejiao Hu[1], Yi Zhu[2], Wendi Ma [2], Bowen Wang[2], Tan Shi[1], Yaoyu Cao[1], Bai-ou Guan[1], Cheng-wei Qiu [4], Yuerui Lu [2,5,6✉] & Xiangping Li [1✉]

The emerging monolayer transition metal dichalcogenides have provided an unprecedented material platform for miniaturized opto-electronic devices with integrated functionalities. Although excitonic light–matter interactions associated with their direct bandgaps have received tremendous research efforts, wavefront engineering is less appreciated due to the suppressed phase accumulation effects resulting from the vanishingly small thicknesses. By introducing loss-assisted singular phase behaviour near the critical coupling point, we demonstrate that integration of monolayer $MoS_2$ on a planar ZnO/Si substrate, approaching the physical thickness limit of the material, enables a π phase jump. Moreover, highly dispersive extinctions of $MoS_2$ further empowers broadband phase regulation and enables binary phase-modulated supercritical lenses manifesting constant sub-diffraction-limited focal spots of 0.7 Airy units (AU) from the blue to yellow wavelength range. Our demonstrations downscaling optical elements to atomic thicknesses open new routes for ultra-compact opto-electronic systems harnessing two-dimensional semiconductor platforms with integrated functionalities.

[1] Guangdong Provincial Key Laboratory of Optical Fiber Sensing and Communications, Institute of Photonics Technology, Jinan University, Guangzhou 510632, China. [2] Research School of Electrical, Energy and Materials Engineering, College of Engineering and Computer Science, the Australian National University, Canberra, ACT 2601, Australia. [3] School of Physics and Optoelectronic Engineering, Ludong University, Yantai 264025, China. [4] Department of Electrical and Computer Engineering, National University of Singapore, 4 Engineering Drive 3, Singapore 117583, Singapore. [5] ARC Centre of Excellence in Future Low-Energy Electronics Technologies (FLEET), ANU node, Canberra, ACT 2601, Australia. [6] Centre for Quantum Computation and Communication Technology, Department of Quantum Science, Research School of Physics and Engineering, The Australian National University, Acton, ACT 2601, Australia. [7] These authors contributed equally: Fei Qin, Boqing Liu. ✉email: yuerui.lu@anu.edu.au; xiangpingli@jnu.edu.cn

The modern information technologies empowered by emerging nanophotonics are centralized on the development of integrated opto-electronic devices with compact footprints and integrated functionalities[1,2]. In this regard, high index semiconductors provide a non-fungible material platform for fusing excitonic light-matter interactions and light field manipulations. Associated with the bandgap characteristic, optical antennas made of high-index dielectrics/semiconductors supporting intriguing Mie resonances and geometrical modes open new avenues for wavefront engineering with ultrathin profiles[3–5]. As such, advancing developments in flat optics have been realized through structuring sub-wavelength high-index dielectric nanoantennas to tailor optical resonances or phase accumulation effects at interfaces to fulfil diverse functionalities of interest, such as metasurfaces[6–8], metaholograms[9,10] and metalenses[11–13]. However, these dielectric elements normally have wavelength-scale thicknesses and high aspect ratios, which represent a daunting challenge in terms of the stringent fabrication precision and deteriorate the compactness of the ultimate devices.

The downscaling of integrated opto-electronic devices to the nanometre scale has shed light on two-dimensional (2D) materials with ultra-thin thicknesses and peculiar optical and electronic properties[14,15]. In particular, the bandgap of transition metal dichalcogenides (TMDs) exhibits a crossover from an indirect bandgap in the bulk state to a direct bandgap in the monolayer limit. Consequently, monolayer TMDs have been intensively investigated for excitonic light-matter interactions associated with direct bandgaps, such as for efficient resonance mirror[16,17], extremely high quantum efficiency in photocurrent generation and the photoluminescence process[18–20]. Their capability for wavefront engineering based on phase accumulation effects is appreciated until very recently as high-index Mie resonators[21] and metalenses[22] at a thickness of hundreds of nanometres. However, such phase accumulation effects are markedly suppressed when the thickness of TMD sheets is decreased to a few atomic layers, leading to insufficient phase modulations far less than $0.3\pi$[23]. To date, integrated strong light field manipulation devices by monolayer materials remain an alluring perspective but largely unattainable yet.

Drastically deviating from conventional phase accumulation effects relying on the real parts of high refractive indices, we theoretically and experimentally reveal that depositing monolayer MoS$_2$ on a properly designed uniform substrate allows a $\pi$ phase shift over a physical thickness of 0.67 nm, leveraging the loss-assisted singular phase. The monolayer TMDs allow us to precisely control the phase modulation with ultra-high spatial uniformity, which is not possible for conventional bulk thin film materials. The spatially uniform and giant phase modulation of $\pi$ arising from 2D monolayers will enable tremendous applications in ultra-thin optical components. As proof-of-principle experiments, binary phase-modulated supercritical lenses with a broadband sub-diffraction-limited focusing capability and meta-holograms have been successfully demonstrated. Our approach allows the realization of 2D flat optical elements with considerable miniaturization and supreme integration capabilities, approaching the physical thickness limit of the material.

## Results

### Loss-assisted phase shift to the extent of the $\pi$ level on a monolayer MoS$_2$ sheet.
In a compound structure composed of monolayer MoS$_2$ on a transparent dielectric layer-Si substrate, an abrupt Heaviside shift in the reflection phase can be readily achieved by judicious parametric design to create a spot of critical coupling[24,25]. By tuning the refractive index ($n$) and thickness ($t$)

of the dielectric layer, an extremely large loss-assisted phase difference in reflected light can be achieved at the wavelength of interest, as depicted in Fig. 1a, where $\emptyset_{\mathrm{MoS}_2}$, $\emptyset_{\mathrm{subs}}$, and $\Delta\Phi$ represent the phase shift of the light reflected from the MoS$_2$ sheet, the dielectric film-coated Si substrate, and their difference, respectively. In such configuration, Si substrate with an infinite thickness prohibits any transmission of light irradiating the system, which makes the compound structure act as a single-port resonator system. Such a single-port model can be well described by temporal coupled mode theory (CMT)[25–27], and the complex reflection coefficient can be derived as:

$$r = -1 + \frac{1/Q_r}{-i(\omega/\omega_0 - 1) + (1/Q_r + 1/Q_a)/2} \qquad (1)$$

where $\omega$ and $\omega_0$ represent the frequency of the incident light and the resonance frequency, respectively, and the quality factors $Q_r$ and $Q_a$ are related to the radiation rate and absorption rate, which mainly depend on the loss property of the interface. The loss-assisted phase shift is determined by the competition between $Q_r$ and $Q_a$. Considering the uniform multilayer configuration, the monolayer MoS$_2$ induced phase difference $\Delta\Phi$ as a function of the refractive index ($n$) and the thickness ($t$) can be readily derived using transfer-matrix methods[28], as plotted in Fig. 1b–c. From this phase diagram, we can see that with an optimal combination of the parameters $n = 1.98$ and $t = 65$ nm, an abrupt Heaviside $\pi$-phase shift in the reflected light can be achieved at the wavelength of 535 nm by a monolayer MoS$_2$ sheet with a thickness of 0.67 nm.

To gain insight into the Heaviside phase jump, complex reflection coefficients of the multilayer systems with and without the MoS$_2$ sheet are calculated using transfer-matrix methods (see Supplementary Note 1) and plotted in Fig. 1d, with a fixed refractive index of $n = 1.98$. As inferred from Eq. (1), the critical coupling state with perfect absorption will occur when $Q_r/Q_a = 1$, corresponding to the singular point of the iso-phase contours marked by the blue star region in Fig. 1c. The over-coupling state of such a multilayer system can be distinguished from the under-coupling state by the critical coupling point. When the system is of lower absorption losses with $Q_r/Q_a < 1$, the system is in an over-coupling state, and the vectorial complex reflection coefficient covers all four quadrants (the olive-green curve in Fig. 1d), which means that the reflection phase can cover the $2\pi$ range. In contrast, when the system has higher absorption losses with $Q_r/Q_a > 1$, the system is in an under-coupling state, and the complex reflection coefficient is bound in the second and third quadrants (the colourful curve in Fig. 1d), which indicates that the reflection phase shift can only undergo a less than $\pi$ phase modulation when sweeping the thickness of the dielectric layer. Through controlling the competition between the dissipation and radiation by tuning the refractive index ($n$) and thickness ($t$) of the dielectric layer, a $\pi$ phase shift difference can be achieved between the over-coupling state without a MoS$_2$ sheet and the under-coupling state with monolayer MoS$_2$ (Supplementary Figs. 1 and 2). Suppose a dielectric material with $n = 1.98$ is used, a phase difference $\Delta\Phi$ (the intersection angle in the phase diagram) of $\pi$ can be reached at the wavelength of interest of 535 nm (light wavelength in our phase-shifting interferometry system, will be explained later), when the dielectric layer has the optimal thickness of 65 nm. A ZnO film can be a perfect candidate to meet the requirements of a high refractive index and transparency in the visible region.

Under the guidance of the above theoretical prediction, we deposited a ZnO thin film with a 65 nm thickness as the dielectric layer on top of a Si substrate by atomic layer deposition (ALD). Then, mechanically exfoliated MoS$_2$ flakes with different

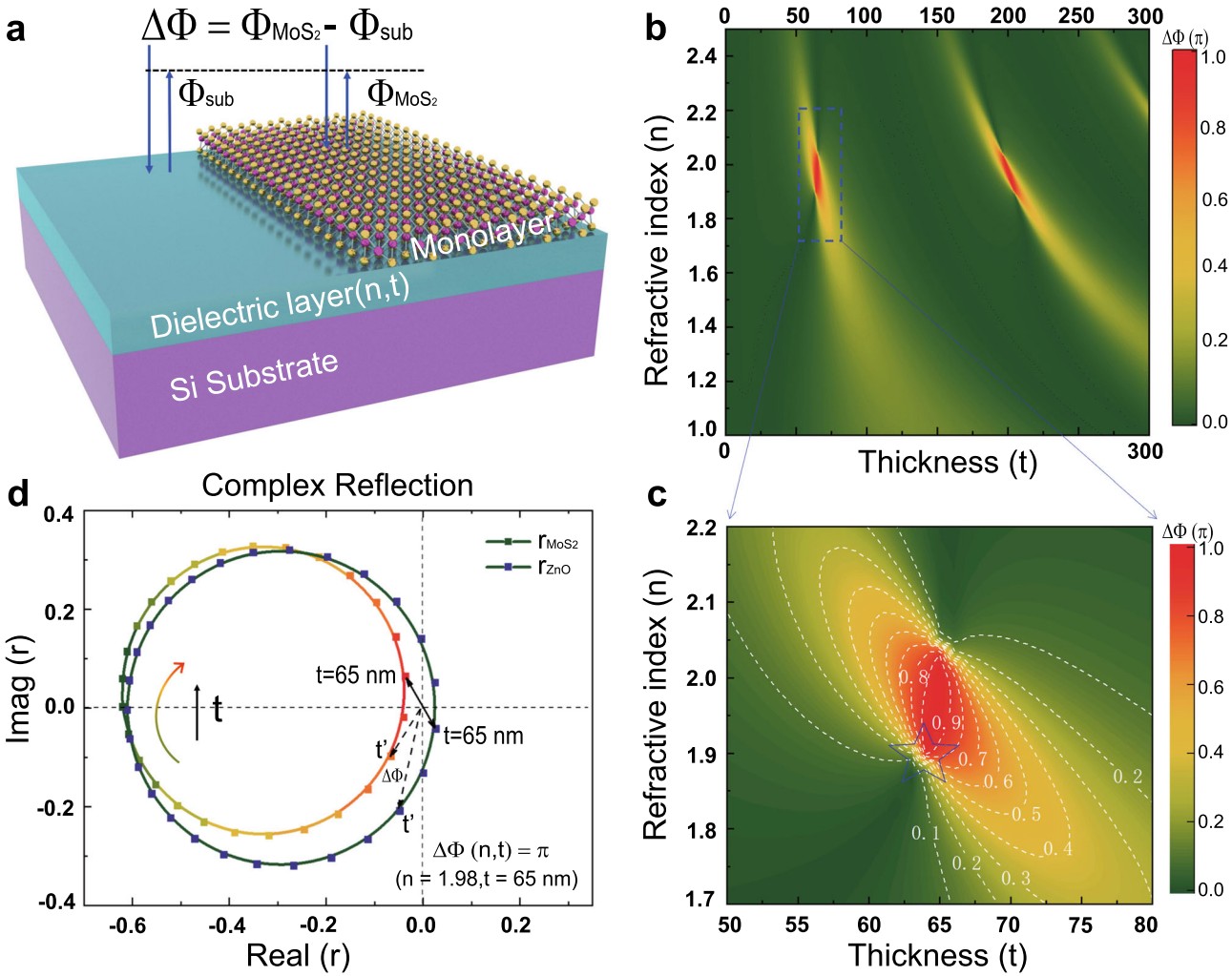

**Fig. 1 Principle of loss-assisted giant phase shift of π arising from a monolayer MoS₂ sheet. a** Schematic illustration of the phase shifts (Δ∅) of the reflected light between a MoS₂ sheet ($\varnothing_{MoS_2}$) and a Si substrate ($\phi_{sub}$) coated by a uniform dielectric film with different refractive indices ($n$) and film thicknesses ($t$). **b** Simulation results of the loss-assisted phase modulation dependence on the thickness ($t$) and refractive index ($n$) of the dielectric layer. **c** Zoom-in view of the blue dashed region in (**b**). White dashed lines are the iso-phase contours. The singular point of the iso-phase contours marked by the blue star represents the critical coupling point at which the system transfers between the over-coupling and under-coupling states, thus leading to an abrupt phase change in the reflected light. **d** Phasor diagrams of the complex reflection coefficient for the cases with (colourful circle) and without (olive green circle) a 0.67 nm-thick MoS₂ sheet on the ZnO-Si substrate when sweeping the thickness of the ZnO sheet from 0 to 130 nm. A π phase difference can be reached at the optimal ZnO film thickness of 65 nm.

thicknesses were transferred onto the ZnO/Si substrate. The layer numbers of the MoS₂ flakes were determined by the optical contrast in microscope images taken before the transfer and further confirmed by Raman spectroscopy data measured from the same samples transferred onto the ZnO/Si substrate (see 'Methods', Supplementary Note 2 and Supplementary Fig. 3). The transferred flakes were measured by the phase-shifting interferometry (PSI) technique[23], and the layer-dependent phase shift with different colours is shown in Fig. 2a. The line profile plotted in Fig. 2b shows that a remarkably high (0.89 ± 0.048) π phase shift arising from the monolayer MoS₂ can be experimentally achieved. This phase shift corresponds to an optical path length of 238 nm, which is approximately 350 times larger than the physical thickness of the monolayer MoS₂. The deviation in the phase shift between the theoretical and experimental results can be largely attributed to the non-ideal refractive index control in the ZnO film deposition process by ALD (Supplementary Fig. 4). In order to further confirm our model, we also measured the phase shifts of 1–4 L MoS₂ samples on Al₂O₃ film with a

measured refractive index ($n$) of 1.65 and an optimized thickness of 78 nm (Supplementary Fig. 5). The experimental data are well-matched with the theoretical predictions (Fig. 2c). By comparison, ZnO with well-fitted refractive index is notably superior to Al₂O₃ as well as other dielectric materials, especially in the monolayer MoS₂ sheet condition, as the simulation and experimental results shown in Fig. 2c. Such a π phase shift makes MoS₂ a good candidate for the construction of binary phase optical devices, for instance, a planar metalens, which is the essential component of miniaturized optical systems and offers a compact design for integrated photonic systems and optoelectronic applications[29–31].

**Monolayer supercritical lens for sub-diffraction-limited focusing.** To date, three different types of planar metalenses, including zone plates[32–36], photon sieves[37,38] and metasurfaces[39,40], have been intensely investigated for optical focusing and imaging. For the construction of these planar lenses, regardless of the class to which they belong, a sufficient phase modulation depth is a

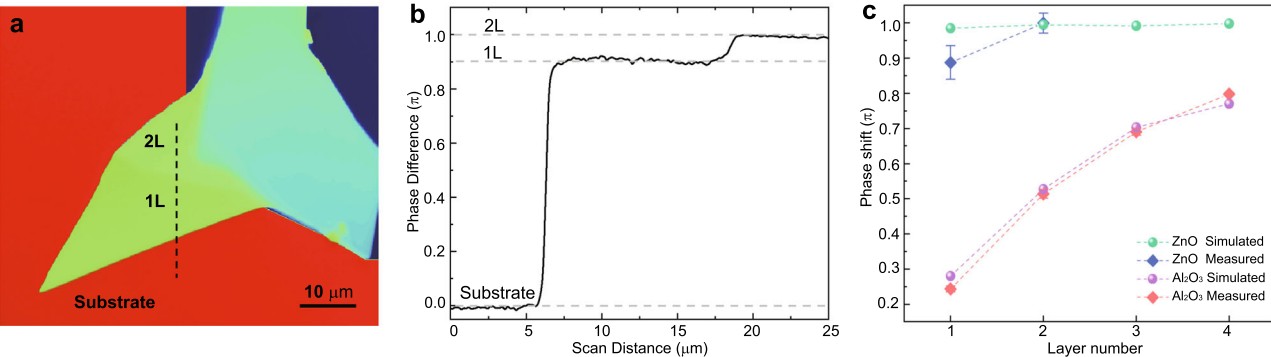

**Fig. 2 Experimental demonstration of a giant π phase shift arising from an atomically thin MoS₂ sheet. a** PSI image of a mechanically exfoliated MoS₂ flake on a ZnO/Si substrate. The thickness of the ZnO layer is 65 nm. Different colours represent different thicknesses of MoS₂ flakes. The areas labelled "1 L" and "2 L" are mono- and bilayer MoS₂, respectively. **b** PSI measured phase difference versus position of 2 L and 1 L MoS₂ along the dashed line marked in (**a**), which depicts that (0.89 ± 0.048) π and (1 ± 0.028) π phase shifts can be achieved for mono- and bilayer MoS₂ sheets on the ZnO/Si substrate, respectively. **c** Statistical data of the phase shift values with error bars from PSI for 1 L, 2 L, 3 L, and 4 L MoS₂ samples on Al₂O₃-coated surfaces and 1 L and 2 L MoS₂ samples on ZnO-coated surfaces. For each layer number of MoS₂, at least five different samples were measured experimentally, and the error bars represent standard deviation of phase shift values in the characterization results, which arises from the inconsistent experimental conditions.

prerequisite. To create π or 2π phase modulation, most of the reported planar lenses have a wavelength-scale thickness, restraining them from use in ultra-compact integrated devices. Supercritical lens (SCL) is a typical representative zone plate-type planar metalenses that can achieve sub-diffraction-limited focusing and imaging in the far field by precisely modulating the interference conditions from each zone belt supporting different spatial components[41–43]. Employing the proposed phase modulation strategy, an atomically thin SCL on a monolayer MoS₂ sheet with a thickness of only 0.67 nm was designed and experimentally demonstrated (see 'Methods' and Supplementary Note 3 and Supplementary Table 1). A large area monolayer MoS₂ sheet was prepared by chemical vapour deposition (CVD) combined with a film transfer process to a ZnO/Si substrate. Then, a direct fs laser scribing technique was applied for the fabrication process (see 'Methods'), as shown in Fig. 3a.

A scanning electron microscopy (SEM) image of the fabricated SCL and a sectional zoom-in view are shown in Fig. 3b, c, which indicate that the fabrication error can be controlled to under ±20 nm to guarantee the focusing performance without significant deviations from the design. An optical image of the fabricated SCL is presented in Supplementary Fig. 6. To confirm the material composition and pattern integrity after the laser process, the fabricated SCL on the monolayer MoS₂ sheet was characterized by the Raman mapping technique. The featured peaks of the $E_{2g}$ and $A_{1g}$ modes, as well as their frequency difference in the Raman spectrum (Supplementary Fig. 3), confirm that the MoS₂ sheet is basically in monolayer state[44]. The complete and sectional lens patterns constructed by plotting the integrated $A_{1g}$ Raman peak intensity of MoS₂ and the Raman peak of the Si substrate are shown in Supplementary Fig. 7, which indicate the fidelity of the SCL pattern after fs laser fabrication.

By using a customized characterization system and illuminating the phase-type SCL with a 535 nm laser beam, a focal spot with a radius of 595 nm was experimentally obtained at a distance of 45 μm from the lens plane, as shown in comparison with the simulation results presented in Fig. 3d, e. Such a focal spot is on the sub-diffraction-limited scale with a lateral size of ∼0.7 Airy units (AU), where AU stands for the Airy spot radius of a diffraction-limited system. The line profile intensity shown in Fig. 3f clearly reveals the agreement between the simulation and experimental results of the lateral size of the focal spot. Moreover, such sub-diffraction-limited focal spot could maintain its

focusing property along the Z-axis, forming an optical needle (Fig. 3g, h). As depicted in Fig. 3i, the focal spot can always keep its radius below the Rayleigh Criterion from z = 40 to 50 μm with stable intensities, which is useful for potential applications in fabrication and information capture. The detailed characterization process is shown in 'Methods' part and Supplementary Fig. 8. The slight deviation in the intensity distribution along the needle region between the experimental and simulation results arises from the fabrication error in the belt width as well as the insufficient phase shift for the belts (Supplementary Note 4 and Supplementary Figs. 9 and 10). Even though the inherent physics of singular phase shift nearby the critical coupling point sets a low reflection efficiency, there are still spaces for further improvement of its efficiency (Supplementary Fig. 11). The experimentally measured focusing efficiency has been given around 1.2% (Supplementary Note 5 and Supplementary Fig. 12). It should be noted that to break the Rayleigh Criterion the principle of supercritical focusing relies on energy re-distribution into the sidelobes in the focal plane as well as the subsidiary foci along the optical axis, fundamentally restricting its focusing efficiency. Nevertheless, our monolayer supercritical lens still exhibits a similar focusing efficiency with literature of sub-diffraction limited metalens at wavelength-scale thicknesses[32,33], and outperforms recently reported monolayer TMD lens with a focusing efficiency of 0.08% by excitonic resonance induced insufficient phase modulation capability[45].

**Broadband focusing with an invariable sub-diffraction limited spot size.** Benefitting from the unique k dispersion properties[46], atomically thin MoS₂ sheets can not only enable a giant phase shift at a single wavelength but also maintain such prominent phase modulation capability over a broadband (Supplementary Figs. 13 and 14). Interestingly, such a broadband phase shift (greater than π/2) can be significantly expanded by increasing the thickness of the MoS₂ sheets from monolayer to bilayer. As shown in Fig. 4a, four critical points arise at different wavelengths, then their adjacent giant phase shift regions are clustered into a plateau in the phase diagram. Thus, a remarkably expanded bandwidth of the phase shift can be obtained in visible frequencies. A high phase shift of greater than π/2 can be achieved with bilayer MoS₂ sheets for the wavelength range from 435 nm to 585 nm, which is reasonably coincident with the abrupt variation region of the k value of MoS₂ sheets, corroborating the loss-assisted phase shift mechanism. In this aspect, MoS₂ transcends

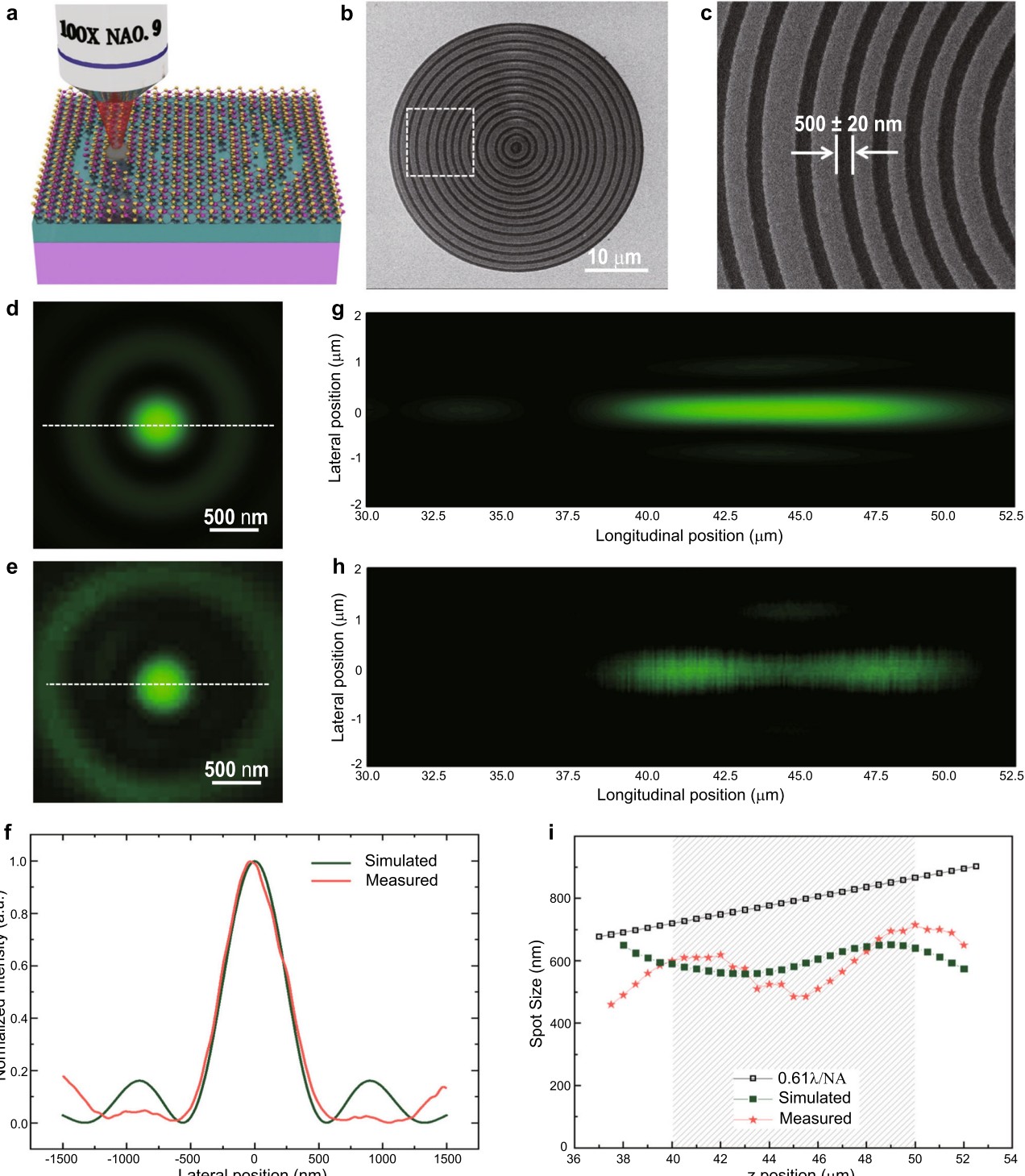

**Fig. 3 Demonstration of a monolayer SCL with sub-diffraction-limited focusing. a** Illustration of a binary phase-type SCL on a monolayer $MoS_2$ sheet obtained by fs laser scribing. **b** SEM image of an SCL on a monolayer CVD-prepared $MoS_2$ sheet. **c** Zoom-in view of the dashed box in (**b**), which depicts that the accuracy of the fs laser scribed belts can be controlled at ±20 nm. **d**, **e** Simulation and measured intensity distribution in the focal plane at a longitudinal position of z = 41 μm, which is the brightest position along the needle in the experimental results. **f** Comparison of the line intensity profiles of the lateral size of focal spots between simulated and experimental results. **g**, **h** Theoretical and measured intensity profiles of the optical needle along the propagation distance in the region between z = 30 μm and 52.5 μm. **i** Comparison between the lateral sizes in the optical needle region and the Rayleigh criterion. The light shadow indicates the optical needle region. Olive green cubes, orange stars, and black open cubes depict the simulated spot size, measured spot size and Rayleigh Criterion, respectively.

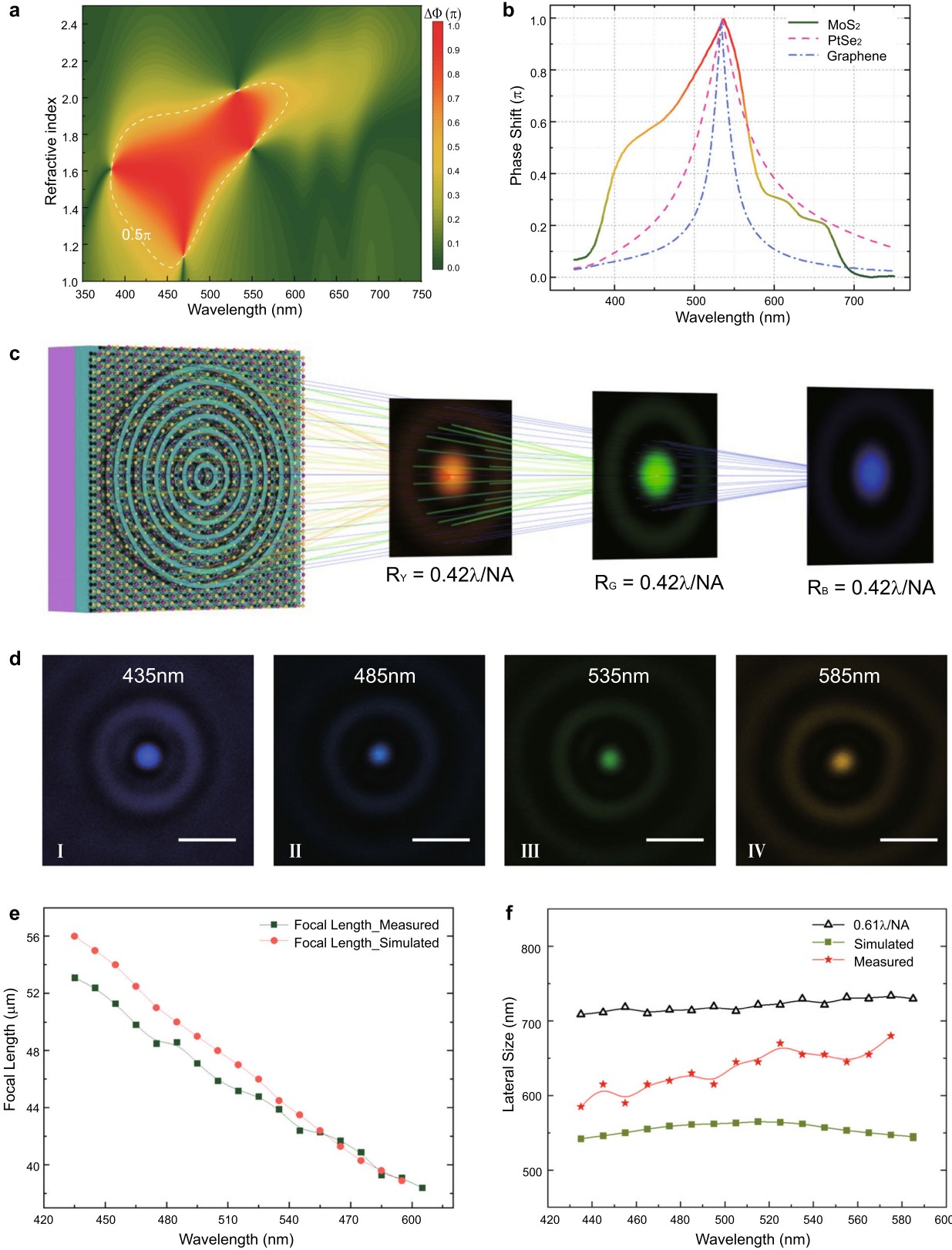

most other 2D materials due to its high refractive index ($n$) and remarkable loss dispersion property ($k$), as shown in the simulation comparison results presented in Fig. 4b.

Such an advantage provides us the possibility to demonstrate an atomically thin broadband SCL, as schematically shown in

Fig. 4c. By using a bilayer $MoS_2$ sheet on a ZnO/Si substrate, a broadband SCL with optimal parameters following a multi-channel optimization algorithm was fabricated (Supplementary Fig. 15). The optical characterization results reveal that sub-diffraction-limited focusing can be achieved over the broadband

**Fig. 4 Atomically thin SCL with broadband sub-diffraction-limited focusing. a** Broadband response of the phase shift for a bilayer $MoS_2$ sheet with a 65 nm ZnO dielectric layer. Multiple transition states arise at different wavelengths due to the large $k$ dispersion property of the $MoS_2$ sheet, which makes the $MoS_2$ sheet ideal for broadband phase shifts. The area inside the white dashed line is the region with phase shifts of $0.5\pi$ and above. **b** Comparison of the phase shift properties between three typical 2D materials. Bilayer $MoS_2$ can achieve a much broader phase shift range than bilayer $PtSe_2$ and four-layer graphene with similar thicknesses. **c** Illustration of broadband sub-diffraction-limited focusing by a $MoS_2$-based SCL, $R_Y$, $R_G$ and $R_B$ represent the focal spots radius of the yellow, green and blue light, respectively. **d** Experimentally measured intensity profile of an atomically thin SCL at the focal plane for selected wavelengths. The complete intensity profiles for more wavelengths are presented in Supplementary Fig. 17. Scale bar: 1000 nm. **e** The broadband SCL shows a negative dispersion, and the focal length has an inverse dependence on the wavelength with good linearity. The orange dot line and olive-green cube line depict the simulated and measured focal lengths for each wavelength. **f** Comparison of the spot size for different wavelengths with the Rayleigh Criterion, which shows that the atomically thin SCL can achieve a sub-diffraction-limited focusing property in the region from blue to yellow. The green cubes, orange stars, and black open triangles depict the simulated spot size, measured spot size and Rayleigh Criterion, respectively.

from 435 nm to 585 nm. Selected measured light intensity profiles for the focal spots with different wavelengths are shown in Fig. 4d (for more results, see Supplementary Figs. 16 and 17). In addition, as Fig. 4e shows, the focal length of the SCL for different wavelengths exhibits great linearity, which is an outstanding advantage for application in wavelength scanning optical tomography imaging with super-resolution capability[47]. As one type of diffractive lens, the SCL exhibits negative dispersion properties, and the lateral size of the focal spots can be much more stable than that of the counterpart refractive lens, since the numerical apertures of the diffractive lens will change with the wavelength following the same trend. In our results, the focal spots of the atomically thin SCL can retain their lateral size at $550 \pm 15$ nm in the simulation and $640 \pm 50$ nm in the experiments from 435 nm to 585 nm (Fig. 4f). Such a feature is distinct from previous results of planar metalens[32,33], which exhibit a strongly dispersive spot size. To push the state-of-the-art, more complicated light field manipulation functionalities including broadband grating deflections and monochromatic holograms were experimentally demonstrated, as shown in Supplementary Note 6 and Supplementary Figs. 18 and 19.

## Discussion

In summary, we theoretically and experimentally demonstrated a phase shift of $\pi$ arising from monolayer $MoS_2$ on a well-designed planar substrate by utilizing the singular phase behaviour near the critical coupling point. Unlike conventional Mie-resonance-based meta-optics, where phase accumulation effects are sensitive to sub-wavelength scale dimensions and geometries, our approach for light field manipulation provides a facile and lithography-free method to develop novel and high-performance optical components with thicknesses approaching the physical limit. To nourish the 2D flat optics, monolayer-thick $MoS_2$ SCL, bilayer broadband SCL and meta-holograms have been demonstrated. The monolayer TMDs with atomic surface roughness provide us fantastic candidate materials to engineer the wavefront with ultra-high spatial uniformity, which cannot be realized using conventional bulk materials due to the intrinsic micro-fabrication variation and errors. Monolayer TMDs with $\pi$ phase modulations may find potential applications in the fields of self-modulating photoluminescence, function-integrated photodetection, exciton field-effect transistor, and even next generation optical computing, etc. Associated with the appealing direct bandgap properties, our demonstration unlocks the full potential of a new class of 2D optics with long-sought integration and miniaturization capabilities and opens a new route to develop photonic integrated circuits incorporating optical wavefront modulators and detectors on the same chip.

## Methods

**The preparation of the $MoS_2$ sheet**. Dielectric materials ZnO was deposited on the doped silicon wafers by using Atomic Layer Deposition (ALD) system (Cambridge Nanotech ALD Fiji F200). Mono-layer and bi-layer $MoS_2$ samples for the characterization of layer-dependent phase shift were prepared by mechanically exfoliation and then transferred onto $SiO_2$/Si chips. The CVD grown chip-scale $MoS_2$ samples were synthesized on $SiO_2$/Si substrates in a hot-wall furnace. Wet transfer method was used to transfer the CVD-grown $MoS_2$ samples onto ZnO/Si substrates for lens making. The detailed preparation and characterization of atomic thin $MoS_2$ is shown in Supplementary Note 2.

**Design of the binary supercritical lens**. The design principle of SCL is based on the Rayleigh-Somerfield diffraction theory combined with the particle swarm optimization algorithm (PSO). The radius of the lens, as well as the numerical apertures, are set on basis of the customized characterization system with R = 40 μm, and NA = 0.4, respectively. For achieving broadband response of the sub-diffraction limited focusing, we have improved the optimization algorithm and established a multi-channel optimization algorithm mechanism, as shown in Supplementary Fig. 15. The target focal spot size has a radius of $0.42\lambda$/NA which in between the Rayleigh Criterion and Super-Oscillation Criterion. The detailed design procedure is shown in Supplementary Note 3.

**Fabrication by fs laser scribing process**. The optimized atomic thin supercritical lens is fabricated by fs laser direct scribing system. The fs pulsed beam with the central wavelength of 780 nm was employed. The repetition rate and pulse width of the laser system are 80 MHz and 80 fs (Chameleon Ultra II). The objective lens NA = 0.9 (Olympus, MPLAN FL N 100X/0.9 BD) was used in the scribing process for focusing the fs laser beam into a spot size with diameter around 500 nm. The $MoS_2$ film is mounted on a piezostage (PI, P-563.3CD), and get the binary phase supercritical lens through the stage scanning. Through precisely control the laser power and scanning speed, the line width with 500 nm on $MoS_2$ films can be readily obtained. The fabricated atomic thin supercritical lens is inspected by the Raman mapping to confirm the fidelity (Renishaw's inVia Raman Microscope), the full structure as well as sectional parts Raman mapping results of the supercritical lens are shown in Supplementary Fig. 7.

**Optical characterization of atomic thin supercritical lens**. The lens characterization is performed by a customized imaging system, as shown in Supplementary Fig. 8. A super-continuum laser (Fianium WL-SC-400-4-PP) is used as the light source. After the light emit out and modulated by the collimator, a planar-convex lens and objective lens with NA = 0.45 (Olympus, 20X) are used in combination to create a plane wave front for illuminating the atomically thin supercritical lens. The alignment between the light beam and the lens is performed by a motorized stage (PI, M26821LNJ). The reflective beam modulated by the atomically thin supercritical lens was collected by the same objective with NA = 0.45 and then the diffraction pattern was acquired by a high magnification imaging system to obtain the XY intensity distribution. The x–z cross section of the intensity distribution is obtained by scanning the supercritical lens along the z direction by the Piezo stage (PI, P-736.ZRN) at a step size of 200 nm, and then map the intensity distribution in the longitudinal plane. Sweeping the wavelength of super-continuum laser, we can get the broadband response of the supercritical lens.

## Data availability

All relevant data can be obtained from the corresponding author under reasonable request.

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

## Acknowledgements

This research was supported by National Key R&D Program of China (2018YFB1107200), National Natural Science Foundation of China (NSFC) (Grant Nos.61705085, 62075085, 61675093, 61705084), Guangdong Provincial Innovation and Entrepreneurship Project (Grant 2016ZT06D081), Guangdong Basic and Applied Basic Research Foundation (Grant No. 2020B1515020058), Guangzhou Science and Technology Program (Grant no. 202002030258), Fundamental Research Funds for the Central Universities (Grant No.21620446). Y.L. acknowledge funding support from National Heart Foundation Australia (Grant No. 102018), Australian Research Council (ARC) Discovery Project (DP180103238) and ARC Centre of Excellence in Future Low-Energy Electronics Technologies (project number CE170100039) and ARC Centre of Excellence in Quantum Computation and Communication Technology (project number CE170100012). We acknowledge the facility support from ACT node of the Australian National Fabrication Facility (ANFF).

## Author contributions

F.Q. and B.L. contributed equally to this work. Y.L. and X.L. conceived the idea and supervised the project. F.Q., L.Z., and W.F. designed the lens structure. F.Q., L.Z., J.L., and T.S. perform the optical characterization. B.L. and Y.Z. fabricated ZnO substrates and conducted optical phase measurements; B.L., W.M., B.W., and Y.L. synthesized the $MoS_2$ films and performed the material characterization. F.Q., D.H., and X.L. performed the modelling and simulation. F.Q., B.L., Y.C., B.G., C.Q., Y.L., and X.L. analysed data and prepared the manuscript. All authors contributed to the discussion.

## Competing interests

The authors declare no competing interests.
