## [Peer Review File · Nature Communications]

REVIEWER COMMENTS

Reviewer #2 (Remarks to the Author):

Authors demonstrate the MoS₂ binary phase-modulated supercritical lenses with sub-diffraction-limited focal spots of 0.7 Airy units (AU) in blue to yellow wavelength range. The principle, method, 2D- materials, and experimental results are not new to most people. Authors need to carefully address why their work can be published by Nature Communications? A few comments to help authors to clarify the importance of their work.

1. To focus light by using 2D material is not a new topic. Many papers have been published, such as Nano Letter 18(11), 6961-6966 (2018), and <https://arxiv.org/ftp/arxiv/papers/1411/1411.6200.pdf>, etc. Previous works demonstrated not only the focusing but also imaging. Please explain why your work is so important? Your manuscripts showed the low quality and weak focusing profile and point spread function, while others showed not only good focusing but also the imaging capability. Please address the uniqueness and innovation in comparison with previously published many articles. By the way, the references of this paper cited many papers from authors but not closely related to the contents of this paper. An updated and proper reference list is needed.

2. Similarly, the phase modulation in the z-axis is not new. There are papers, such as Communications Physics 2, 156, 2019, etc., that demonstrated by metalens or zone plates, their phase modulation can cover the whole 2π . However, in this manuscript, there is only 1π phase modulation, which would cause some problems for the optical functionalities.

3. The feasibility of the applications is an important issue as well.

4. The results (Fig. S19) of meta-holograms on the atomic thin MoS₂ sheet are not good enough, which have a low contrast. The author should find out the cause of this problem and improve the results before re-submission. The current results in the supplementary material seem to appear the authors prepared in a hurry.

My recommendation is that authors should make a major revision of this paper. An updated and proper reference list is needed.

Reviewer #3 (Remarks to the Author):

I enjoyed reading the updated version of this manuscript. The authors have taken care of my comments in a satisfactory manner. After the removal of the interesting, but quite preliminary photoluminescence results, I am now happy to recommend publication of this important work. I think the idea of using a single (few) atomic layer(s) to strongly manipulate the reflection phase will inspire a wave on new experiments. In my opinion it is still too early to compare the performance of the elements against other dielectric flat optics (e.g. diffractive optical elements or Mie-resonant metasurfaces) in terms of the performance. These elements had much more time to develop and the

performance and application areas of these unique, conceptually new devices is expected to improve over time.

Responses to the reviewers' comments

Reviewer #2 (Remarks to the Author):

Authors demonstrate the MoS₂ binary phase-modulated supercritical lenses with sub-diffraction-limited focal spots of 0.7 Airy units (AU) in blue to yellow wavelength range. The principle, method, 2D- materials, and experimental results are not new to most people. Authors need to carefully address why their work can be published by Nature Communications? A few comments to help authors to clarify the importance of their work.

Reply: We thank the reviewer for reviewing our manuscript again. To clearly illustrate the novelty and advances of our work, we have summarized as follows.

Firstly, the high refractive indices of transition metal dichalcogenide (TMD) 2D materials are appreciated for light field manipulation applications until very recently [Refs. 21-22 in our revised manuscript]; however, these samples prepared in hundreds-of-nanometers thickness to access Mie resonances and phase accumulations are conventionally perceived essential. When the thickness of 2D materials decreases to atomic monolayers, approaching the physical thickness limit, the common light phase modulation methods such as Mie resonances and phase accumulations are not applicable any more. By utilizing widely perceived adverse effect of losses in nanophotonics, we demonstrate that the integration of a monolayer MoS₂ sheet with only 0.67 nm thickness on a uniform substrate can create the spot of the critical coupling and hence for a remarkable π phase shift, which represents the world's first of its kind. The underlying physics makes it fundamentally different from previous reports.

Secondly, in addition to the demonstration of the nontrivial phase jump crossing the critical coupling point at a single wavelength, we discovered that the highly dispersive extinctions of MoS₂ sheets empowers a prominent broadband phase shift (greater than $\pi/2$) by increasing the thickness of the MoS₂ sheets from monolayer to bilayer. To showcase the capability in broadband light field manipulations, a variety of meta-optics have also been successfully demonstrated, including broadband beam deflection by meta-grating and broadband meta-holograms, etc.

Thirdly, utilizing a facile direct laser writing method, we demonstrate binary phase modulated meta-optics based on atomically thin surface corrugations, representing the thinnest planar optics. In specific, we demonstrate atomically thick supercritical lens that allows to break the diffraction limit for super-resolved focusing, which stands for its own strength and makes it fundamentally different from the previous demonstration of diffraction-limited FZP lens on 2D materials.

Overall, our work reports a remarkable π phase shift in monolayer 2D materials approaching the physical thickness limit of materials and demonstrates its potential applications in light field manipulation meta-optics. We believe our manuscript demonstrates a timely and important advance at the merge of 2D materials and nano-photonics, and should meet the criteria of *Nature Communications*.

1. To focus light by using 2D material is not a new topic. Many papers have been published, such as Nano Letter 18 (11) 6961-6966(2018), and <https://arxiv.org/ftp/arxiv/papers/1411/1411.6200.pdf>, etc. Previous works demonstrated not only the focusing but also imaging. Please explain why your work is so important? Your manuscripts showed the low quality and weak focusing profile and point spread function, while others showed not only good focusing but also the imaging capability. Please address the uniqueness and innovation in comparison with previously published many articles.

Reply: We thank the reviewer for suggesting relevant literatures. Though demonstration of light focusing by thick 2D materials is not new, to achieve remarkable phase modulation by monolayer 2D materials remains still elusive. The most notable difference that distinguishes our work from previous published results is that a remarkable π phase modulation is achieved with monolayer TMDs with sub-nanometer thickness, while others achieve π or 2π phase modulation usually by many layers of 2D materials with thicknesses of tens to hundreds nanometer. The two references mentioned by the reviewer which are cited as Refs. 22 and 23 in our revised manuscript demonstrate light focusing with a thickness of 190 nm and 6.28 nm, respectively. Thus, it is less comparable with our demonstration of phase modulations by monolayer MoS₂ with a thickness of 0.67 nm. Downscaling the TMDs sheet from multilayer to monolayer will bring a crossover from indirect bandgap to direct bandgap property of the TMDs, then make it a good candidate for excitonic applications such as the photocurrent generation and photoluminescence process. Our demonstration unlocks the full potential of a new class of 2D optics with long-sought integration and miniaturization capabilities and opens a new route to develop photonic integrated circuits incorporating optical wavefront modulators and detectors on the same chip.

By the way, the references of this paper cited many papers from authors but not closely related to the contents of this paper. An updated and proper reference list is needed.

Reply: We have followed the reviewer's suggestion and re-organized the reference list. Only the papers closely related to this works are cited.

2. Similarly, the phase modulation in the z-axis is not new. There are papers, such as Communications Physics 2, 156, 2019, etc., that demonstrated by metalens or zone plates, their

phase modulation can cover the whole 2π . However, in this manuscript, there is only 1π phase modulation, which would cause some problems for the optical functionalities.

Reply: We thank the reviewer for professional comments. Indeed, there are many published works on metalens and zone plate based on phase modulations. However, the phase modulation strength with 1π or 2π is usually realized by dielectric resonators with a thickness in the scale close to the wavelength. As shown in the paper mentioned by the reviewers (Ref 8 in the revised manuscript), a dielectric metasurface based zone plate is experimentally demonstrated with higher performance than conventional amplitude-modulated zone plates. However, 2π phase modulation for the wavelength of 635 nm in this work is achieved through the electric and magnetic dipole resonance of α -Si nanorods with a thickness of 330 nm. By contrast, in our work, we demonstrated the remarkable 1π phase shift with monolayer MoS₂ approaching the physical thickness limit of the material, by utilizing widely perceived adverse effect of losses in nanophotonics. Since only single layer MoS₂ is employed, binary phase contrast of 1π from the scribed and un-scribed regions can be achieved on the uniform substrate. Nevertheless, such a binary phase modulation is sufficient and widely adopted for the construction of many flat optical device with binary configuration. To illustrate capabilities in complex light field manipulations, we have demonstrated planar metalens, metagrating, and holographic imaging, etc,

3. The feasibility of the applications is an important issue as well.

Reply: In this work, we successfully demonstrated a remarkable π phase modulation with monolayer MoS₂. The configuration of “semiconductor substrate / dielectric buffer layer / monolayer TMD film” we used in this work is fully compatible with reported TMD devices with opto-electronic functions. Efficient phase modulation capability will release the full potential of monolayer TMDs, could pave the route for 2D flat optical elements approaching the physical thickness limit with considerable miniaturization and supreme integration capabilities.

Combining with the wavefront manipulation, direct bandgap and tunable excitonic property of monolayer TMDs, we envision its bright future in integrated optical circuits. It will significantly benefit many fundamental investigations including exciton-polariton interactions, information valleytronics, and nonlinear optics, and also may find potential applications in the fields of self-modulating photoluminescence, function-integrated photodetection, exciton field-effect transistor, atomic thin augmented/virtual reality system, even the next generation computing, etc.

Finally, although quite a few photonic devices have been demonstrated on monolayer MoS₂, advancing progresses for the integration system still require significant efforts in terms of cooperation of the researchers among the material synthesis, device fabrication, system

architecture design in the future. The reviewer #3 also shares the same opinion in this aspect as “it is still too early to compare the performance of the elements against other dielectric flat optics (e.g. diffractive optical elements or Mie-resonant metasurfaces) in terms of the performance. These elements had much more time to develop and the performance and application areas of these unique, conceptually new devices is expected to improve over time”

4. The results (Fig. S19) of meta-holograms on the atomic thin MoS₂ sheet are not good enough, which have a low contrast. The author should find out the cause of this problem and improve the results before re-submission. The current results in the supplementary material seem to appear the authors prepared in a hurry.

Reply: We thank the reviewer for this valuable suggestion to further improve the quality of our manuscript. In regards to the low-quality meta-holographic results shown in Fig S19, it comes from the relative low pixel numbers in the hologram pattern with only 400*400 pixels. If a larger hologram pattern is generated, the quality can be significantly improved. To verify that, a new meta-hologram with 1000*1000 pixels is generated with the same algorithm and patterned on the atomically thin MoS₂ sheet by fs laser scribing system. High fidelity holographic images have been clearly demonstrated experimentally, as shown in Figure S19 in the revised supplementary materials and the panel B in Figure R1 in below.

Figure R1 Comparison of reconstructed holographic images between patterns with 400*400 pixels in panel A (the results in previous manuscript) and 1000*1000 pixels in panel B (in the revised manuscripts).

My recommendation is that authors should make a major revision of this paper. An updated and proper reference list is needed.

Reply: We deeply appreciate the reviewer for professional comments and valuable suggestions. In response to those constructive suggestions, we have performed additional experiments on high quality meta-holograms. Brief discussions about the potential applications of such monolayer TMD system with dramatic phase modulation are added. The references are re-organized to make sure that citations are closely related to this work.

Reviewer #3 (Remarks to the Author):

I enjoyed reading the updated version of this manuscript. The authors have taken care of my comments in a satisfactory manner. After the removal of the interesting, but quite preliminary photoluminescence results, I am now happy to recommend publication of this important work. I think the idea of using a single (few) atomic layer(s) to strongly manipulate the reflection phase will inspire a wave on new experiments. In my opinion it is still too early to compare the performance of the elements against other dielectric flat optics (e.g. diffractive optical elements or Mie-resonant metasurfaces) in terms of the performance. These elements had much more time to develop and the performance and application areas of these unique, conceptually new devices is expected to improve over time.

Reply: We thank the reviewer for reviewing our manuscript again, and recommending the publication of our work in *Nature Communications*. We share the same opinion with the reviewer that it is not the time to compare the performance of such atomic thin elements with other dielectric flat optics, joining the competition with other diffractive optical elements is not the main focus of this work. The novelty of this work is to demonstrate giant phase modulation capability on monolayer 2D materials by utilizing widely perceived adverse effect of losses in nano-photonics, and provided an exciting potential of a new class of 2D optics with long-sought integration and miniaturization capability. Advancing progresses for the integration system still require significant efforts in the aspect of material synthesis, device fabrication, and system architecture design in future.

REVIEWERS' COMMENTS

Reviewer #2 (Remarks to the Author):

Authors demonstrate the MoS₂ binary phase-modulated supercritical lenses with sub-diffraction-limited focal spots of 0.7 Airy units (AU) in blue to yellow wavelength range. The principle, method, 2D- materials, and experimental results are not new to most people. However, the authors keep insisting that they demonstrated the integration of a monolayer MoS₂ sheet with only 0.67 nm thickness on a uniform substrate can create the spot of the critical coupling and hence for a remarkable π phase shift, which represents the world's first of its kind. Judging from their responses to my previous comments, and their efforts on a new meta-hologram with 1000*1000 pixels on the atomically thin MoS₂ sheet by fs laser scribing system, as shown in Figure S19 in the revised supplementary materials, the revised manuscripts can be accepted now.

Responses to the reviewer's comments

Manuscript ID: NCOMMS-20-26426A

Manuscript title: “ π -phase modulated monolayer supercritical lens”

Reviewer #2 (Remarks to the Author):

Authors demonstrate the MoS₂ binary phase-modulated supercritical lenses with sub-diffraction-limited focal spots of 0.7 Airy units (AU) in blue to yellow wavelength range. The principle, method, 2D- materials, and experimental results are not new to most people. However, the authors keep insisting that they demonstrated the integration of a monolayer MoS₂ sheet with only 0.67 nm thickness on a uniform substrate can create the spot of the critical coupling and hence for a remarkable π phase shift, which represents the world's first of its kind. Judging from their responses to my previous comments, and their efforts on a new meta-hologram with 1000*1000 pixels on the atomically thin MoS₂ sheet by fs laser scribing system, as shown in Figure S19 in the revised supplementary materials, the revised manuscripts can be accepted now.

Reply:

We thank the reviewer for reviewing our manuscript again, and recommending the acceptance of our work in *Nature Communications*. We are also very grateful for all the comments in the past few round review, which are very important and valuable to improve the quality of our manuscript.